# Developing Techniques for Closed-Loop-Recycling Soda-Lime Glass Fines through Robotic Deposition

**Maria Sparre-Petersen [1],* and Simona Hnídková [2]**

1 The Royal Danish Academy—Architecture, Design, Conservation, 2100 Copenhagen, Denmark
2 Technical University of Denmark, 2800 Lyngby, Denmark; simona.hnidkova1@gmail.com
* Correspondence: msp@kglakademi.dk

**Abstract:** Glass is made from sand—a finite resource. Hence, there is a need to maintain glass in the industrial cycle as described in the Ellen MacArthur Foundation's circular-economy diagram. This research project examines the reallocation of material resources in the form of waste glass fines from the industrial recycling process for soda-lime glass. According to the plant manager of Reiling Glasrecycling Danmark ApS, the fines are currently sold to be used for insulation. Although this process prolongs the lifespan of the fines before they become landfill waste, a closed-loop circular option would be preferable. In order to establish a closed-loop circular model for waste glass fines, this research investigates their material and aesthetic qualities and proposes a strategy for maintaining the fines in the closed loop cycle together with the soda-lime glass. The fines are manipulated through robotic deposition and formed into 3D geometries. To expand the aesthetic applications for the material, an investigation is conducted by combining 3D geometries with the traditional glassmaking techniques of glassblowing and casting. The research contributes knowledge of the materials' technical qualities including printability, durability and workability of the 3D prints combined with cast or blown recycled container glass as well as with blown waste glass fines. Technical obstacles are revealed and alternative routes for further explorations are suggested. Finally, the performative and aesthetic qualities of the results are discussed, while artistic applications for recycled soda-lime glass fines remain to be explored in future research.

**Keywords:** artistic glass; 3D printed waste glass fines; closed-loop-recycling; sustainability; sustainable development goals; circular economy



## 1. Introduction

Glass is a versatile material used in applications ranging from perfume bottles to medical instruments to spacecrafts, etc. Shortages in virgin materials (UNEP 2019) points to the urgency of recycling as much glass as possible. This is in line with the UN Sustainable Development Goals no. 9 Industry, Innovation and Infrastructure and no. 12 Responsible Consumption and Production (United Nations 2015). The largest category of industrial glass today is container glass with a 37% share of the global market (Furszyfer Del Rio et al. 2022). Most container glass is made from the soda-lime composition and all soda-lime glass can be recycled together at certain temperatures.

We presume that engaging and experimenting with recycled glass materials can generate explicit technical and aesthetic knowledge as well as tacit knowledge (Polanyi 1966), which can contribute to the development of a new sustainable technical and material foundation for artistic glass making. This could potentially influence the footprint of artistic glass through a practice that is more sensitive to and appreciative of the recycled materials and their inherent value.

## 2. Maintaining Recycled Soda-Lime Glass Fines in the Closed-Loop Industrial Cycle

As a result of crushing recycled glass containers into shards, the container glass recycling industry produces a residual material in the form of soda-lime glass fines. The

fines are a powder similar to the color powders used by art glass makers, although of a different composition (see Figure 1). Presently, the reuse of this material mainly follows a downcycling model as "glassphalt" or insulation (Blicher-Nordkvist 2022), materials that are usually not recycled at the end of their product life.

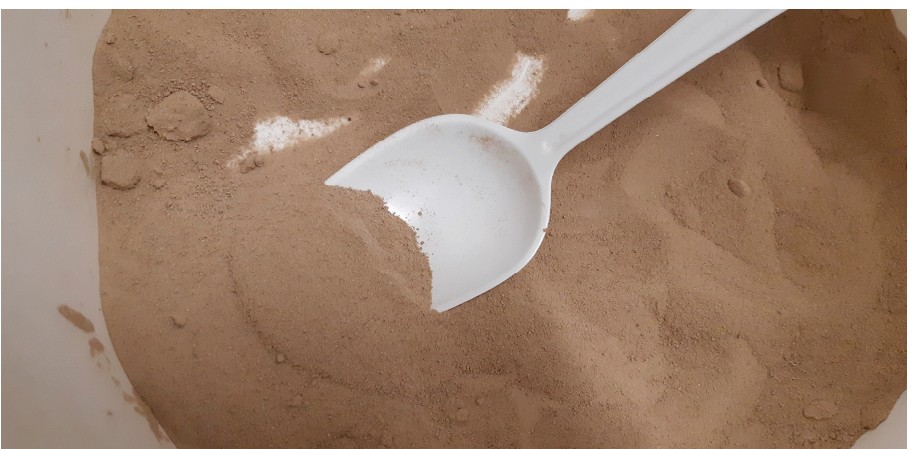

**Figure 1.** Recycled soda-lime glass fines.

This text reports on a research project investigating the plausibility of maintaining recycled soda-lime glass fines (RSGF) in the closed-loop industrial cycle as suggested in the butterfly diagram by the Ellen MacArthur Foundation (2023). By 3D printing the RSGF into solid objects, it is possible to recycle it together with regular recycled container glass. The container glass recycling loop is closed because glass can be recycled endlessly without a loss of material properties (Dyer 2014). Consequently, anywhere the recycling of waste container glass is common practice, the RSGF could be available for glass artists to utilize and maintain in the loop.

Based on artistic experimental material explorations, the research has multiple aims. The initial intent is to find a recipe for the extrusion of the RSGF, which produces 3D geometries that become form-stabile objects following a thermal sintering process. The second is to combine the 3D prints with traditional glass-making techniques of casting and blowing glass, to develop more strategies for utilizing the printing technique. The third is to investigate 3D as well as color options for the printed material.

Hence, we ask the following:

How may experimental artistic material research contribute to sustainable development and aesthetic innovation within the artistic glass field through robotic deposition of recycled soda-lime glass fines?

### 3. State of the Art for Research into Robotic Deposition of Recycled Soda-Lime Glass Fines

Several research projects are exploring the utilization of RSGF through additive manufacturing by introducing it into polymers, where the circular model is obstructed (Ting et al. 2021). We were able to locate only a few projects that are currently exploring alternative use of the fines to establish closed-loop-recycling for RSGF (UTS 2022; Thomsen et al. 2020).

Three-dimensional printing of the RSGF with the use of robotic deposition has been explored in a previous research project by Thomsen et al. (2020), identifying challenges in the outcomes with regards to brittleness and form stability after the thermal sintering process. Thomsen et al. mixed the RSGF with flour and water to produce a substance that was extrudable and maintained the 3D geometry through drying and firing. The flour caused the development of a foamy structure of the material during firing presumably due to the flour burning away. Hence, the options for forming the material were limited as well as the functionality and usability of the results (Thomsen et al. 2020).

This research also utilizes the robotic deposition technique and seeks to remedy the technical obstacles through a set of material tests directed at finding out how different material compositions influence the stability of the results of the 3D robotic deposition of RSGF after thermal sintering at 970 °C. Subsequently, it investigates artistic applications for the RSGF through four series of experiments. In the first series, the RSGF is printed into 3D geometries and then added to cast recycled container glass blocks as well as printed directly on to the blocks. The second series investigates attaching 3D prints to recycled container glass using the blowing technique. The third series explores attaching the 3D prints to blown RSGF and the fourth looks at 3D printing options for RSGF as well as adding colors to the mixture.

## 4. Printing RSFG

To print the glass fines, we use a UR5e robotic arm (see Figure 2) carrying a tube with 1.3 L of the RSGF material mixed with an additional material. The material mixture is extruded by using air pressure, guaranteeing a constant stream of 2.0 bar. The plunger in the tube enables a consistent and smooth extrusion of the material mixture. The tube is connected to the ViscoTec high-precision dosing technology, which allows for the extrusion of the material mixture in a variety of predetermined geometrical shapes. Due to the constant 3 mm, thickness of material deposition, varied nozzle widths and template of design parameters, we are able to observe changes in material characteristics, contributing to the knowledge of how 3D printed glass fines react to forming procedures.

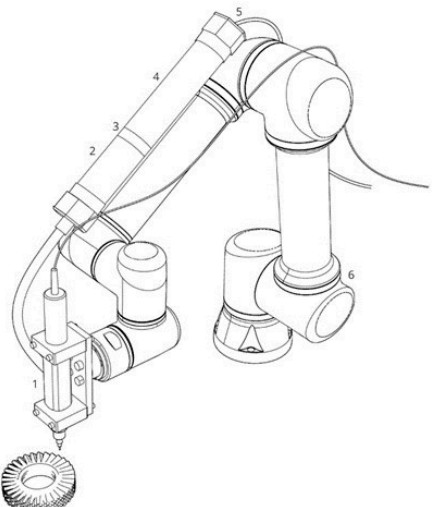

**Figure 2.** 1: ViscoTec, 2: Material mixture, 3: Plunger, 4: Plastic tube, 5: Air pressure source, 6: UR5e.

## 5. Testing Material Composition

As mentioned previously, the RSGF must be mixed with an additional material to enable extrusion through the bores of the UR5e printing setup, tube parts and ViscoTec dosing technology. The additional materials explored in this research are sugar and gelatin, water glass, dispex, Titebond glue and bentonite. The different additional materials affect how the RSFG reacts to different temperatures, how malleable the material is upon reheating, how long it retains heat, how the color is affected, how much it sticks to forming tools, etc.—all information that informs the possibilities for application. Each material is mixed with the RSGF to gain a viscous texture slightly thicker than toothpaste and then extruded into the following samples: two diamond shapes, a cylindrical shape, four lines of a differing thickness and a circle (see Figure 3). Each set is manually extruded using a test syringe of the same thickness as the ViscoTec dosing technology.

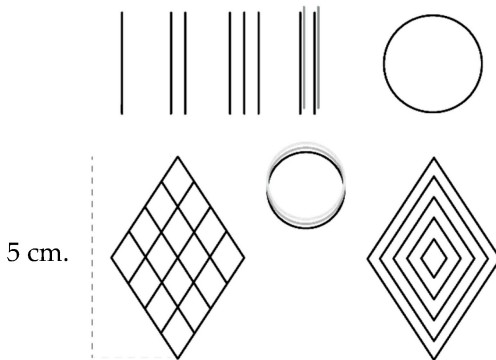

**Figure 3.** Printing diagram for material tests.

The material tests are fired in a ceramic kiln (see Table 1):

**Table 1.** Firing program for material tests.

|  | Time | Temperature | Holding Time |
| --- | --- | --- | --- |
| Initial heat | 3 h | 570 °C | - |
| Rapid heat | 10 min | 970 °C | 10 min |
| Rapid cool | 10 min | 564 °C | End |

To determine which materials to pursue for the following experiments, the additional materials' qualities and performance are analyzed using the following criteria with marks from 1 to 5, where 5 is the best (see Table 2):

**Table 2.** Material quality and performance of additives.

| Material | Extrudability/ Printability | 3D Structure | Performance during Drying | Performance after Firing |
| --- | --- | --- | --- | --- |
| Sugar | 5 | 2 | 4 | 1 |
| Gelatin | 1 | 4 | 3 | 2 |
| Dispex | 3 | 2 | 4 | 4 |
| Water glass | 5 | 1 | 4 | 4 |
| Bentonite | 3 | 4 | 4 | 3 |
| Titebond | 4 | 5 | 4 | 4 |

Extrudability/printability

- cloaking or clumping during extrusion;
- time interval for binding of the layers;
- sensitivity to changing environmental conditions (humidity, temperature).

Ability to form 3D structures

- binding of layers;
- stability during print process;
- possible overhang of layers.

Performance during drying process

- degree of slumping;
- shrinkage;
- cracking;
- deformation.

Material performance after firing at 970 °C

- porosity;

- strength;
- glass look;
- translucency.

The results of the tests showed that sugar performs best with regards to printability while this additive performs moderately with regards to the 3D structure and shows the lowest performance after firing (see Figure 4).

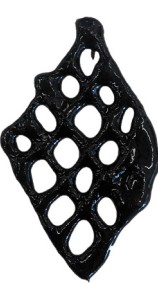 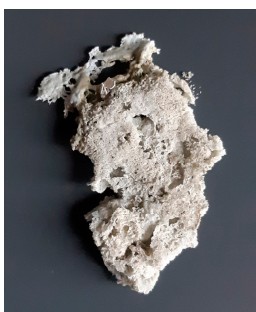

**Figure 4.** Sugar and RSGF before and after firing at 970 °C.

Gelatin, bentonite and Titebond glue perform best with regards to the ability to form 3D structures, while gelatin and bentonite become porous and brittle after firing (see Figures 5–7).

With regards to performance during drying, there is a marginal difference between the materials, indicating that this test parameter was less important than expected.

Water glass and Dispex perform best regarding performance after firing in terms of glass look and translucency, while they perform badly with regards to forming a 3D structure (see Figures 8 and 9).

Based on the test results, Titebond glue was found to be the best overall performing material regarding printability, 3D ability and performance after firing. The outcome is a material that is durable, lustrous and not porous. Titebond glue is therefore selected to be the material additive used in the following experiments.

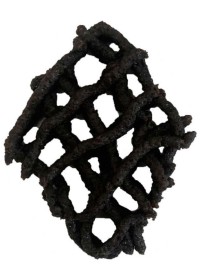 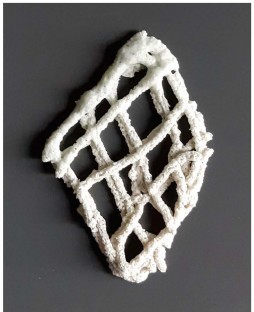

**Figure 5.** Gelatin and RSGF before and after firing at 970 °C.

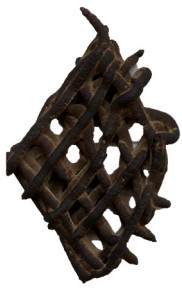 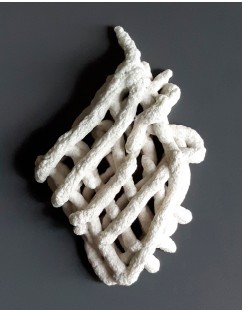

**Figure 6.** Bentonite and RSGF before and after firing at 970 °C.

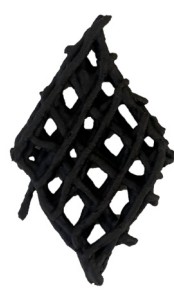
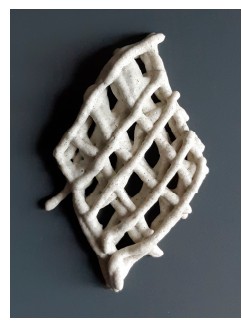

**Figure 7.** Titebond glue and RSGF before and after firing at 970 °C.

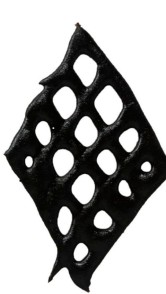
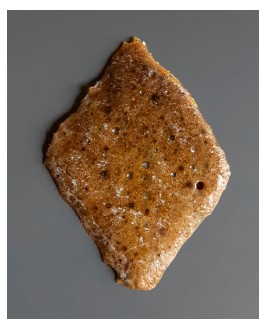

**Figure 8.** Water glass and RSGF before and after firing at 970 °C.

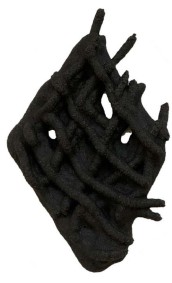
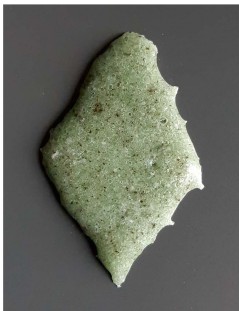

**Figure 9.** Dispex and RSGF before and after firing at 970 °C.

## 6. Techniques for Artistic Applications of the 3D Printed RSGF

Having determined a material additive for aesthetic investigation, we continue to explore and discuss four techniques for the artistic application of the 3D printed RSGF:

- attaching the 3D prints to prefabricated cast recycled container glass blanks in a cold application process followed by thermal sintering;
- attaching the 3D prints to blown recycled container glass in a hot process;
- attaching the 3D prints to blown RSGF in a hot process;
- 3D printing of RSGF, including the addition of ceramic stains.

The 3D modeling software Rhinoceros and the plugin Grasshopper are used to design the 3D geometries (see Figures 10–12) and to translate the models into toolpaths for the UR5e robot (see Figure 13).

The Grasshopper script also enables the control of the printing speed and the layer height, which needs to be calibrated according to the toolpath in order to produce the desired 3D printed shape.

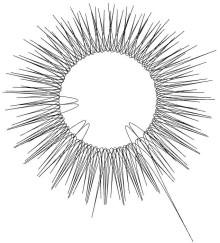

**Figure 10.** Rhino file—top view.

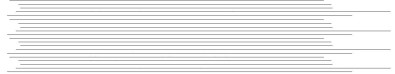

**Figure 11.** Rhino file—side view.

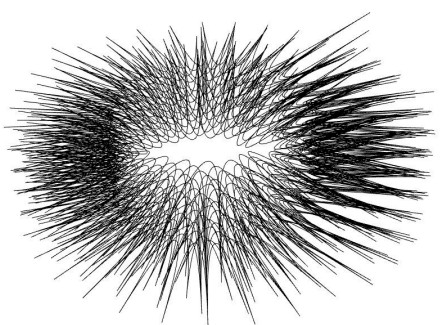

**Figure 12.** Rhino file—perspective.

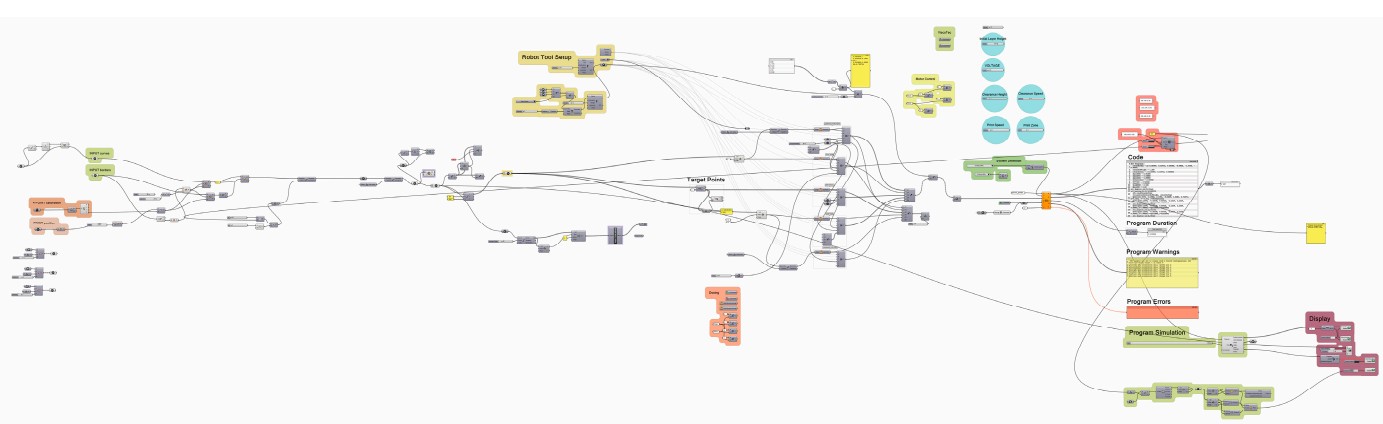

**Figure 13.** Grasshopper script.

*6.1. 3D Printed RSGF Objects Combined with Clear Cast Recycled Container Glass*

An initial series of three firing tests are conducted to determine how the prints adapt to the cast container glass. For this experiment, rectangular glass objects are cast from molten recycled container glass and annealed to even out stresses in the glass. The material mixture made from RSGF with Titebond glue is printed flat and then applied to the cast glass objects (see Figures 14 and 15).

After drying the objects with the 3D prints, they are fired at three different temperatures: 780 °C, 785 °C and 800 °C (see Figures 16–18).

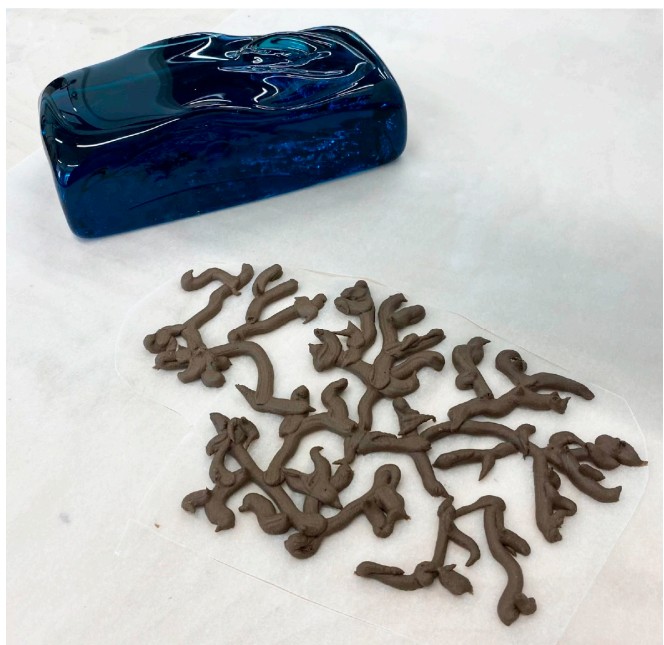

**Figure 14.** 3D print and cast glass object.

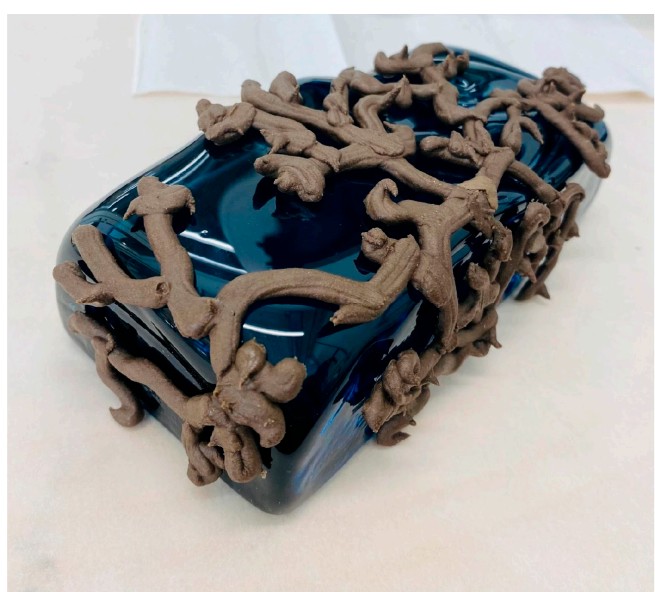

**Figure 15.** Cast glass object with 3D print before firing.

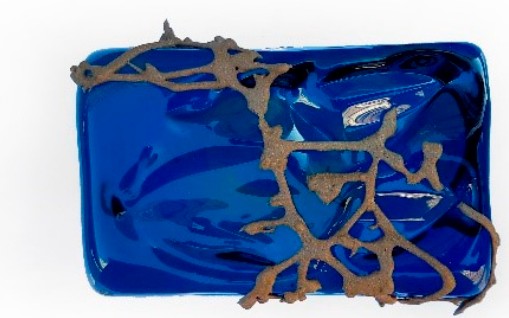

**Figure 16.** Cast container glass with 3D print fired at 780 °C.

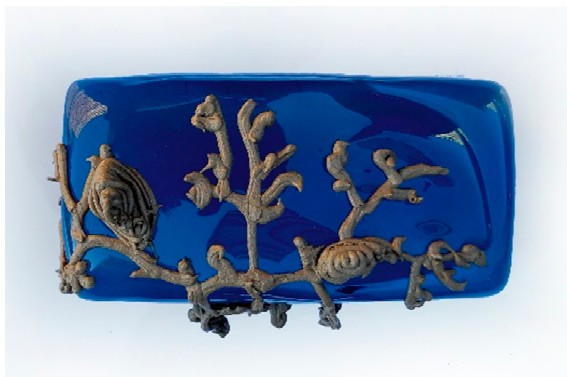

**Figure 17.** Cast container glass with 3D print fired at 785 °C.

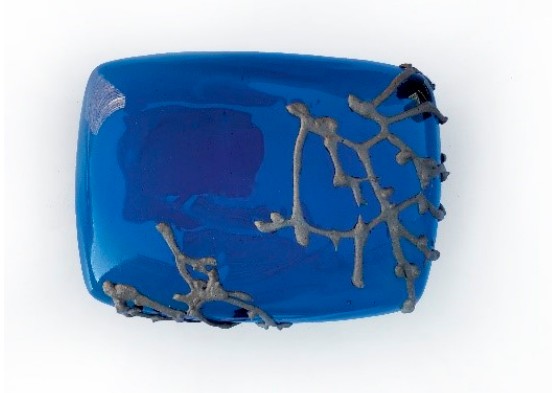

**Figure 18.** Cast container glass with 3D print fired at 800 °C.

The results of the firing tests show that the original shape of the clear glass object is affected by the temperature at 785 °C and 800 °C whereas the 3D printed parts are affected similarly at all three temperatures. These parameters can be adjusted to fit the aesthetic criteria of a given project. The resulting artefacts show no immediate signs of compatibility issues or problems with shrinkage. But after a few months, cracks emerge where the two materials are fused together thoroughly.

The following experiment expands on the 3D ability of the technique as well as on the aesthetic possibilities. Four cast recycled container glass objects are placed on a surface of approximately 35 by 40 cm. The RSGF/glue mixture is printed onto and in-between the four objects in an organic geometry (see Figures 19 and 20). The joined materials are fired according to the firing and annealing program in Table 3:

**Table 3.** Firing and annealing program for cast soda-lime glass blocks with 3D printed RSGF.

|  | Time | Temperature | Holding Time |
| --- | --- | --- | --- |
| Initial heat | 10 h | 600 °C | - |
| Rapid heat | skip | 785 °C | 10 min |
| Rapid cool | skip | 564 °C | 5 h |
| Anneal drop | 15 h | 420 | - |
| End |  |  |  |

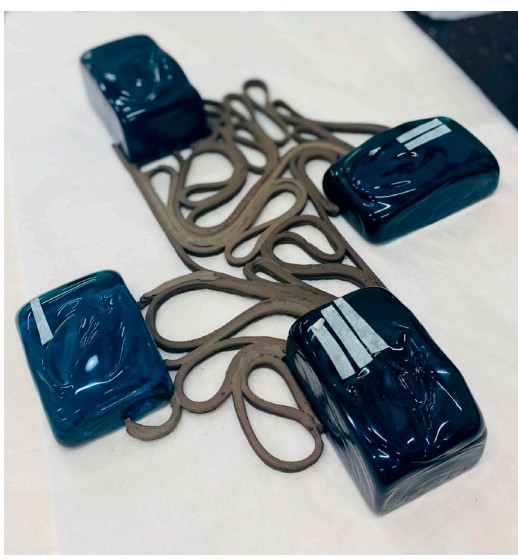

**Figure 19.** Cast container glass with 3D print (process).

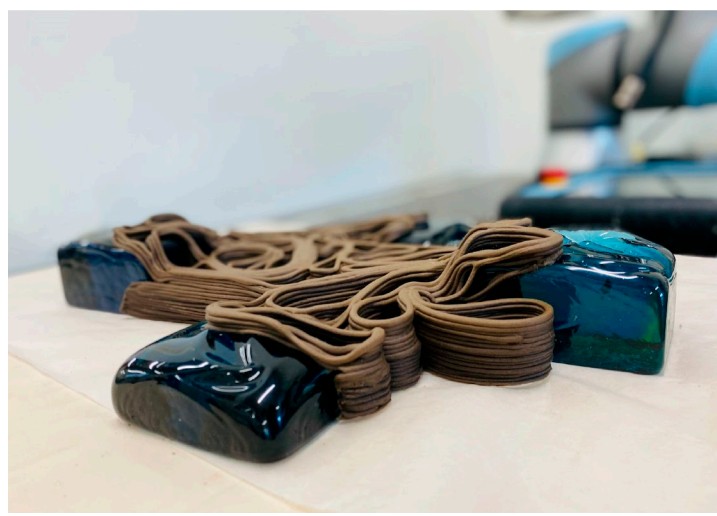

**Figure 20.** Cast container glass with 3D print before firing.

6.1.1. Technical Material Analysis of RSGF Combined with Cast Recycled Container Glass

Combination of the 3D print with the cast objects brings out important qualities of material composition. However, the 3D printed material contracts and changes its form by 15–20% during the drying and firing processes. This phenomenon causes large cracks in the 3D print when applied to larger structures (see Figure 21), rendering this forming strategy less viable for scaling, and for diversifying the aesthetic opportunities. Further explorations into this technique might consider bisque firing the 3D print and using a cold adhesive to combine the prints with cast glass elements.

Returning to the previous temperature tests, it seems odd that the printed material did not reveal indications of shrinkage issues on the individual cast recycled container glass objects. Two differences in the application process may have influenced this outcome: the initial tests were printed in relatively thin layers compared to the larger experiment, and the initial tests were applied after the printing process as opposed to the larger experiment that was printed directly on to the cast objects.

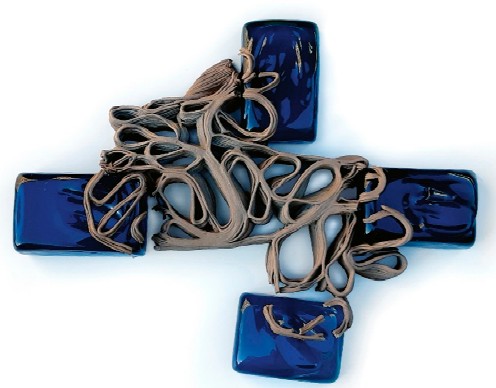

**Figure 21.** Cast container glass with 3D print after sintering at 785 °C.

### 6.1.2. Aesthetical Analysis of RSGF Combined with Cast Recycled Container Glass

Aesthetically, the experiment reveals several possibilities of contrasting the matte RSGF with the transparent/translucent cast glass, combining color and elaborating on form, which might be explored further in future research, through cold adhesion techniques.

### *6.2. 3D Printed RSGF Combined with Transparent Blown Recycled Container Glass*

For the next experiments, 3D printed RSGF is combined with transparent blown recycled container glass. A series of geometries are printed at a suitable size to be attached to a blown molten glass bubble in a process called a "pick-up". The prints are pre-fired in a ceramic kiln at 850 °C (see Table 4)—a temperature determined through a series of previous experiments to be ideal for sintering.

**Table 4.** Firing program for 3D printed geometries for pick-ups.

|  | Time | Temperature | Holding Time |
|---|---|---|---|
| Initial heat | 8 h | 600 °C | - |
| Rapid heat | skip | 850 °C | until pick-up |
| End |  |  |  |

The recycled container glass is melted in a furnace at 1250 °C and is blown into a bubble. When the blown bubble has the desired form and temperature, the kiln with the 3D print is opened and the bubble is gently pressed onto the print—a process called a pick-up in glassblowing terms. This causes the recycled container glass and the RSGF to fuse together (see Figure 22) and the temperature of the entire object is then homogenized by reheating it in the electric melting furnace, which provides a homogenous heat distribution.

The fused object is then removed from the blowpipe and placed in a third kiln called the annealing kiln. It is soaked for a while at a temperature of 564 °C—the annealing point of soda-lime glass—and then cooled down slowly to 420 °C, which is called the annealing process (see Table 5). The annealing process is conducted to even out stresses in the object.

**Table 5.** Annealing program for blown soda-lime glass bubbles with 3D printed RSGF.

|  | Time | Temperature | Holding Time |
|---|---|---|---|
| Soak | 4 h | 564 °C | - |
| Annealing cool | 8 h | 420 °C | - |
| End |  |  |  |

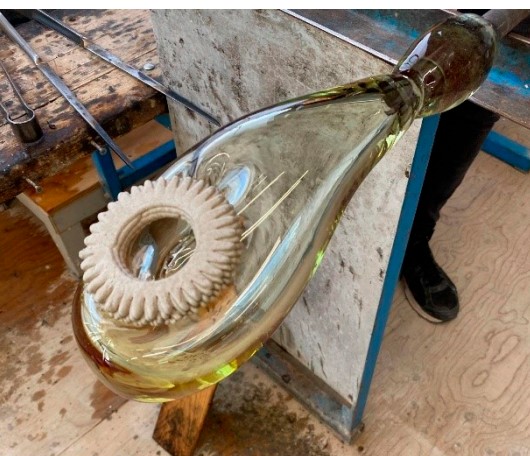

**Figure 22.** 3D printed RSGF attached to blown recycled container glass.

6.2.1. Technical Material Analysis of RSGF Combined with Blown Recycled Container Glass

During the initial firing, the 3D printed RSGF goes through a process called devitrification; this means that the material changes from an amorphous to a crystalline molecular structure. All glass devitrifies at a particular temperature interval, a feature that most glass makers regret, and soda-lime glass devitrifies at a much faster pace than other types of glass (Hodkin and Cousen 1925). A devitrified material has different material qualities than molten glass. Glass has an amorphous molecular structure, whereas a devitrified material has a crystalline molecular structure. This means that the coefficient of expansion of the two materials are not the same, which causes incompatibility between them. This triggers cracks in the blown glass surface where it is in contact with the 3D printed material (see Figure 23).

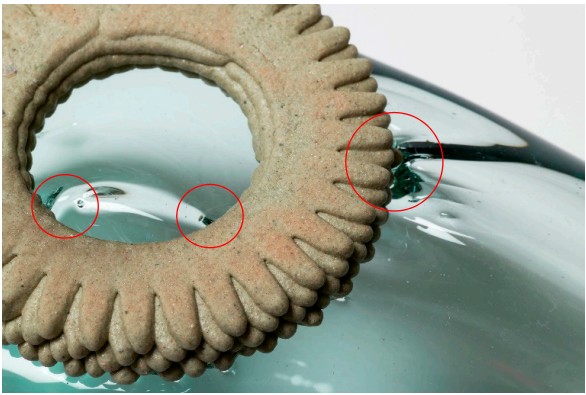

**Figure 23.** Cracks caused by incompatibility.

6.2.2. Aesthetic Analysis of RSGF Combined with Blown Recycled Container Glass

With regard to the aesthetic outcomes of the experiments, the initial idea of combining the 3D printed objects with blown molten glass objects was to wrap the prints around the bubble. This idea proved impossible. The prints were simply too rigid to wrap, and instead just attached to the bubble (see Figure 24). This allows for different aesthetic explorations that could be pursued further in future research given that a method for adjusting the coefficient of expansion was found. We have chosen to explore melting the RSGF rather than pursuing to align the coefficient of expansion of the two materials.

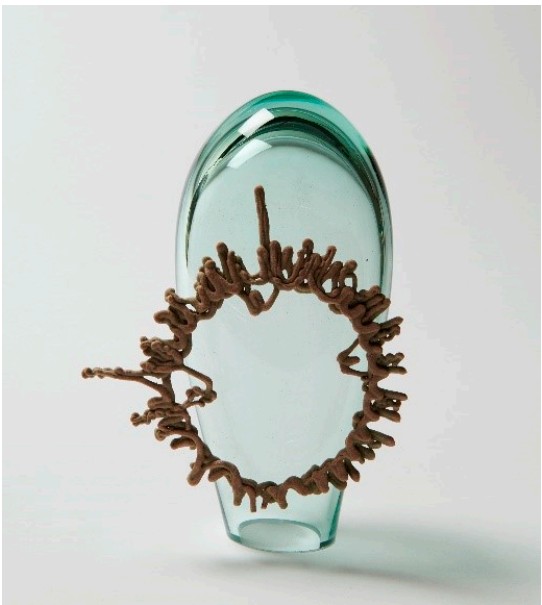

**Figure 24.** 3D printed RSGF on blown container glass.

*6.3. 3D printed RSGF Combined with Blown RSGF*

In the third experiment, the 3D printed RSGF is combined with blown molten RSGF. An initial test is made to find out if the RSGF can be melted into glass, and to feel what the molten material is like. The RSGF is melted in the furnace in the same way as the regular recycled container glass, at 1250 °C. After 2 days of cooking, and stirring with a potato on a stick intermittently, the material turns into a workable glass with very few tiny bubbles (see Figure 25).

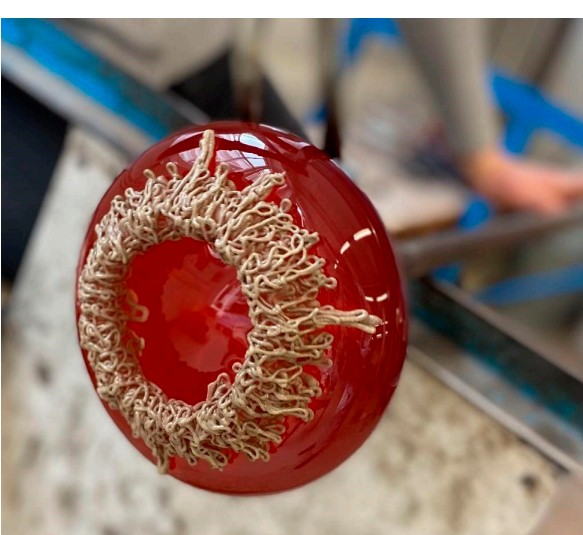

**Figure 25.** 3D printed RSGF on blown RSGF (process).

The RSGF contains non-glass contaminates from the process of crushing and sorting the containers at the recycling plant (Flood et al. 2018). These particles influence the melting process and the resulting molten glass material. The result is a glass that retains the heat longer than the recycled container glass, making it easier to manipulate on the blowing iron. The longer melting process requires an increased energy consumption, which is counterproductive to the sustainable agenda. This must be taken into consideration in relation to the benefits of keeping the glass in the circular loop.

For the experiments, the RSGF is blown into the desired shape. The 3D printed object that has been preheated in a separate kiln is brought out of the kiln and placed on a fireproof plate from which it is attached to the blown shape. After a thorough reheating, the fused object is annealed as described in Section 6.2.

### 6.3.1. Technical Material Analysis of 3D Printed RSGF Combined with Blown RSGF

Our conjecture is that the coefficients of the expansion of the blown and the printed RSGF are different and will cause issues with incompatibility in these experiments just as in the previous experiments. The blown RSGF is too dark to conduct a polariscope test to confirm the conjecture. Therefore, the results of the experiments require a further analysis before conclusions can be drawn.

### 6.3.2. Aesthetic Analysis of 3D Printed RSGF Combined with Blown RSGF

The blown RSGF has a dark greenish brown color that turns black when the glass is thick. The shiny surface of the blown RSGF contrasts the light brown color of the 3D printed part, which allows a range of aesthetic options that can be investigated further in future research if the compatibility test should turn out to be positive against our conjecture (see Figures 26–28).

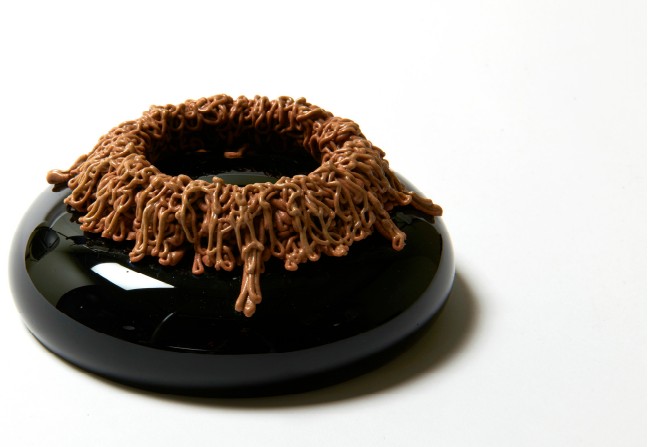

**Figure 26.** 3D printed RSGF on blown RSGF (outcome).

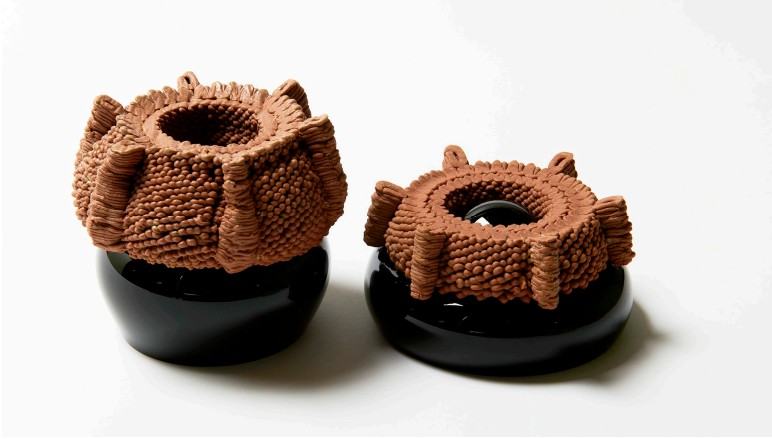

**Figure 27.** 3D printed RSGF on blown RSGF (outcome).

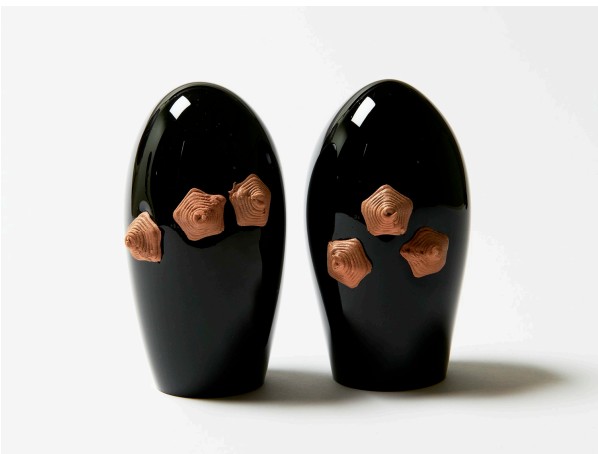

**Figure 28.** 3D printed RSGF on blown RSGF (outcome).

The dark tint of the blown RSGF offers possibilities as a colorant for recycled container glass and might even improve the workability of this material. Additionally, it offers aesthetic opportunities as a colored glass option in and of itself.

*6.4. 3D Printing and Coloring Options for RSGF*

3D printed structures initially produced for the blowing experiments reveal promising results for further explorations of form and aesthetics. Investigations carried out in this research include altering the settings on the robot to pursue different numbers of layers with asymmetrical layouts (see Figures 29 and 30), varying the layout of the extruded lines (see Figures 31 and 32) and printing thicker walls combined with supportive structures and different numbers of layers to test the material's ability to form 3 dimensional objects (see Figures 33 and 34).

Another option for expanding the aesthetic options for the printed material involves adding ceramic stains to the RSGF to enable a wider color palette (see Figures 35 and 36). Ceramic stains are made from fritted and ground oxides, and are usually utilized for dyeing ceramic bodies, engobes or glazes. For this research, *Universalcolor Red 2742* from the company Cerama was utilized.

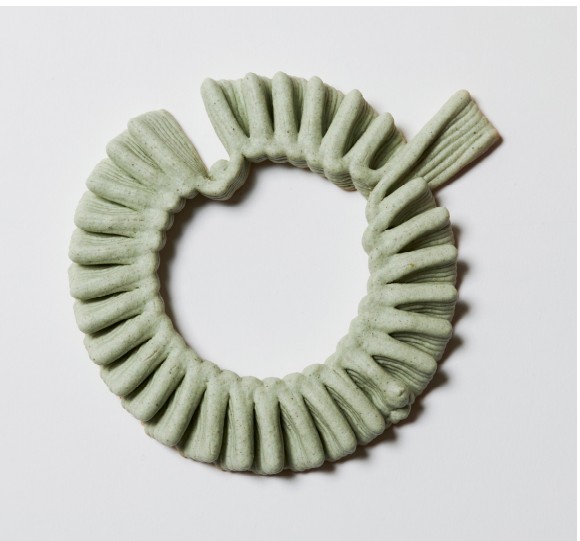

**Figure 29.** Asymmetrical 3D printed RSGF with 7 layers.

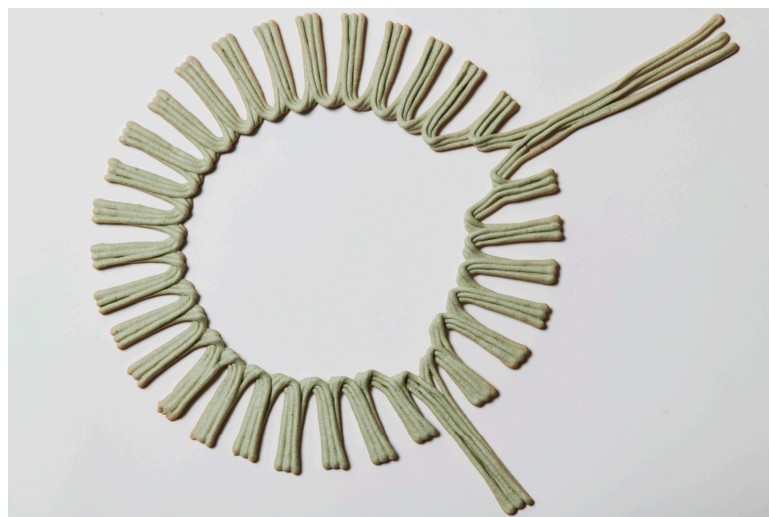

**Figure 30.** Asymmetrical 3D printed RSGF with 3 layers.

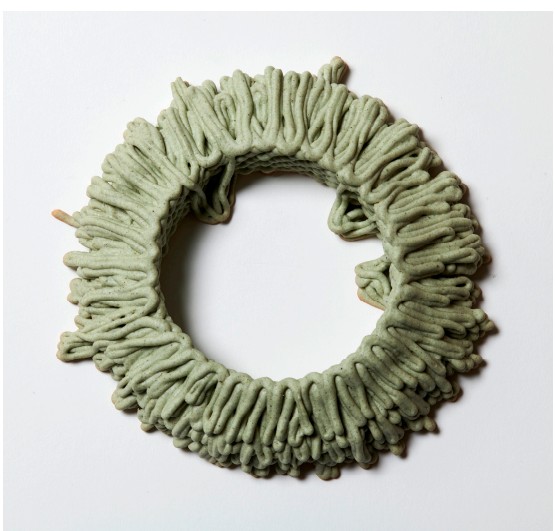

**Figure 31.** 3D printed RSGF with straight lines.

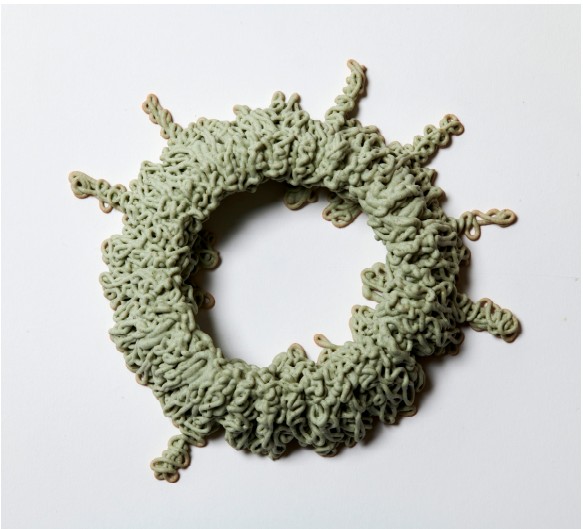

**Figure 32.** 3D printed RSGF with curly lines.

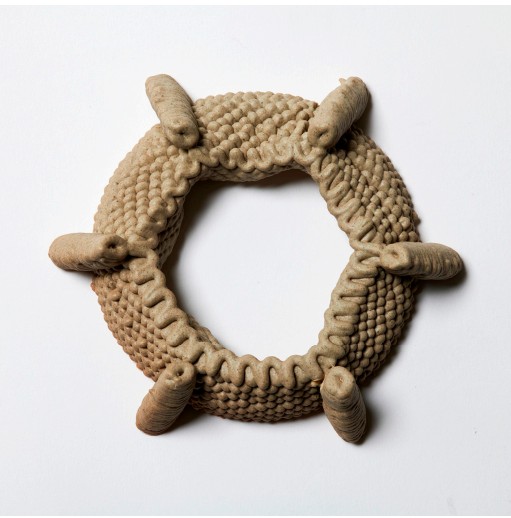

**Figure 33.** 3D printed RSGF object with thick walls and multiple layers intact after firing.

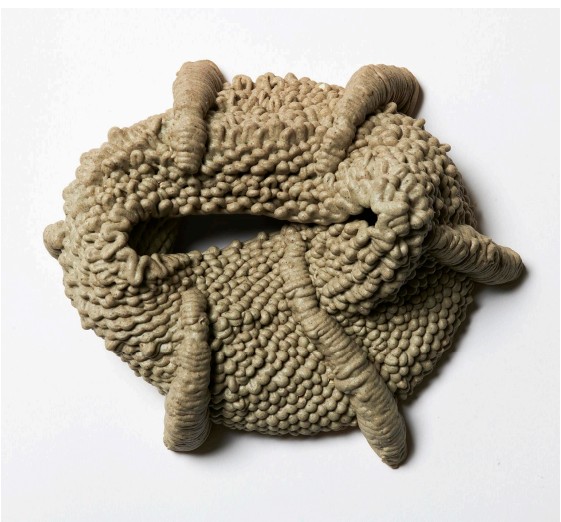

**Figure 34.** 3D printed RSGF object with thick walls and multiple layers collapsed after firing.

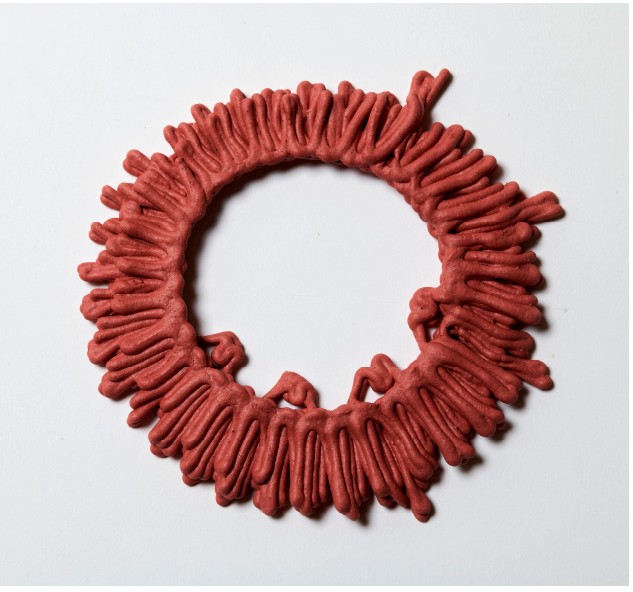

**Figure 35.** 3D printed concentric RSGF geometry with *Universalcolor Red 2742*.

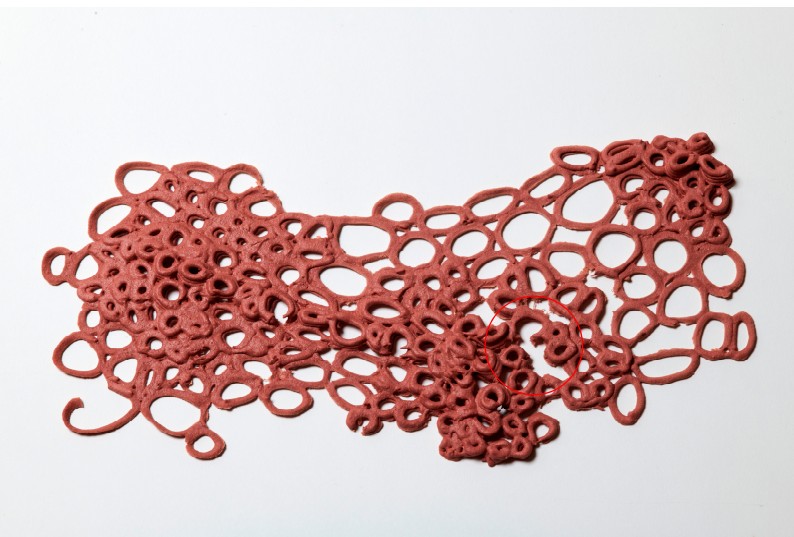

**Figure 36.** 3D printed RSGF geometry of varying numbers of layers with *Universalcolor Red 2742*.

### 6.4.1. Technical Material Analysis of 3D Printed and Colored RSGF

The results of these investigations reveal the most stable three-dimensional results for geometries of a larger width. Adjustments of the speed of the extrusion offer curly lines as an alternative to straight lines. Adding a ceramic stain offers the possibility of expanding the palette of colors. Objects of multiple layers may gain an improved stability by adding external support structures as part of the 3D geometry although limitations to height and width are also influenced by the sintering temperature as revealed in Figure 34. Thin lines and variation in numbers of layers cause the material to tear itself apart during the thermal sintering process, as revealed in Figure 36.

### 6.4.2. Aesthetic Analysis of 3D Printed and Colored RSGF

Aesthetically, the 3D prints show promising results for artistic exploration. Further investigations including the elaboration of support structures, more complex 3D geometries, larger objects and more options for controlling the printing line would be possible directions to pursue. The expanded color palette allows for a more versatile vocabulary for artists to employ in their expression of artistic content.

## 7. Sustainable Development of Artistic Glass through Robotic Deposition of RSGF

This research proposes 3D printing as a strategy for maintaining RSGF in the closed-loop recycling system that is already established for container glass in several countries worldwide. The experimental approach of the research focuses on an environmental aspect of sustainability.

For glass artists to be able to benefit from the 3D printing technique, practice, exercise and implicit material understanding are key in a similar way as they are key to creating through traditional form-giving techniques (Hansen and Falin 2016). Three-dimensional printing brings out the qualitative learning of additive manufacturing such as the extrudability of the material and material ability to form 3D structures. This knowledge can be harnessed to generate more complex 3D structures and, slowly, the technique can potentially develop into a more refined stage. Ford and Despeisse (2016) describe additive manufacturing as a broad range of production technologies that fabricate products layer-by-layer, enabling 3D objects to be 'printed' on demand. Additive manufacturing methods are widely explored presumably because they are less wasteful than traditional subtractive production methods, eliminating molds as well as subtractive waste. Moreover, this technique allows sustainable benefits such as value chain reconfiguration and resource efficiency improvement. Value chain reconfiguration includes using localized production—here, in the form of the locally sourced RSGF—and resource efficiency is improved by

keeping the material in the closed loop rather than downcycling or landfilling it. Hence, in the experimental activities conducted in this research, we investigated 3D printing as a new glass-making technique with sustainable benefits. Three-dimensional printing allows for forming a material into shapes that are unique to this technique. The thermal sintering enables a lower energy consumption than re-melting of the material (Thomsen et al. 2020). The material composition of the results allows for returning the material into the closed loop cycle and additionally the technique offers new aesthetic opportunities for artists to use towards expressing artistic concepts and ideas.

The option of adding ceramic stains as a colorant for a 3D printed material further expands the aesthetic opportunities for the material, although ceramic stains are made from oxides that are most often mined and therefore raise new questions with regard to environmental sustainability. Consequently, further investigations might include research into using other industrial waste materials for coloring 3D printed materials, as suggested by Kucerenkaide (Ignorance Is Bliss 2023), who develops dyes for coloring textiles, ceramic and glass based on industrial byproducts such as waste iron from the water supply industry.

## 8. Method: Experimental Material Research for Sustainable Development

The reported research has employed an explorative experimental attitude towards materials. This enables a maker to develop a deep material sensibility and tacit knowledge of material properties and workability through continuous trials and errors. This way of developing new knowledge has also been identified as a bottom-up approach by Pajunen et al. (2013), which means that the goals of the explorations are not preconceived. Instead, the explorations are led by curiosity and a systematic collection of data based on experiments. Furthermore, in accordance with the network theory as accounted for by Latour (1996) the results of the experiments shape the sensibility of the maker in a reciprocal relationship where the final products express the materials' shaping of the maker as much as the maker is shaping the material. This process has also been described as a "reflection in action" by Schön (1991) as a dialogue with the material, where the material "speaks back" to the maker, which enables a deeper understanding of a problem and its possible solutions. Hence, the trials and particularly the errors of this investigation expand our research group's combined tacit knowledge of the material as well as of the robotic deposition technique. Through the experimental method, a foundation is established for the further development of our own as well as others' explorations of the RSGF. Robots continuously become more and more advanced and easier for non-experts to operate, which will make it easier to explore aesthetic directions for 3D printed RSGF. Hence, other artists may be inspired to try out the technique and add to the knowledge base. This will generate more diverse applications for the material and thereby contribute to the development of more sustainable form-giving options for glass artists.

## 9. Conclusions

Global energy challenges, climate change and geopolitical issues derived from limited sand resources will force the glass field to re-think the standard linear production modes and begin to expand the existing recycling systems to meet future demands of glass products. Based on the results of this research, we suggest a circular economy for RSGF, which may contribute to an adaptation to a circular economy for artistic glass through applications of knowledge from both traditional glass-making techniques and new additive manufacturing.

Traditional artistic glass making is based on significant tacit and methodic knowledge of the material qualities and technical and aesthetical opportunities of glass. This knowledge has been developed through highly specialized artistic glass-making practices since the dawn of the studio glass movement around the middle of the 20th century (Lynggaard 1998). Meanwhile, the glass industry has become more focused on how to streamline productions and develop new technological solutions to be able to increase revenue and meet market demands. Through this research, a new connection between traditional artistic

glass, additive manufacturing and the waste glass industry is proposed that can harness tacit material knowledge of artistic glass making combined with additive manufacturing processes. This can contribute to the sustainable development of the artistic glass field by providing tacit and explicit material knowledge that can lead to new closed-loop-recyclable works of art.

While several of the initial ideas for aesthetical explorations of RSGF proved difficult due to technical issues, the research is contributing to diversifying the aesthetic options for RSGF through the following:

- development of technical material knowledge of how to solve issues of 3D printing RSGF into resilient form-stabile objects;
- development of new tacit knowledge of the processes involved in the production of geometries that are more resilient to firing treatment;
- enabling 3D printing of a material into forms that would be difficult to achieve in any other technique;
- expanding the color range for recycled container glass for artistic applications.

Furthermore, the project is contributing two sustainable options for RSGF:

- 3D printing followed by sintering at 970 °C allows for a lower energy consumption;
- blowing as well as 3D printing allows for returning RSGF into the circular model for recycled container glass.

Hence, the overall results of the research are revealing promising routes for the further exploration of a circular model for RSGF.

**Author Contributions:** Conceptualization, M.S.-P.; methodology, M.S.-P., S.H.; software, S.H.; validation, M.S.-P., S.H.; formal analysis, M.S.-P., S.H.; investigation, M.S.-P., S.H.; resources, not applicable to this article; data curation, not applicable to this article; writing—original draft preparation, M.S.-P., S.H.; writing—review and editing, M.S.-P.; visualization, M.S.-P., S.H.; supervision, not applicable to this article; project administration, M.S.-P.; funding acquisition, M.S.-P. All authors have read and agreed to the published version of the manuscript.

**Funding:** This research was funded by The Royal Danish Academy—Architecture, Design, Conservation and The Danish Arts Foundation, Journal number SKDP 15.2021-0308. Recycled container glass fines were donated by Reiling Glasrecycling Danmark ApS, Universalcolor Red 2742 was donated by Cerama High Temperature Products A/S.

**Data Availability Statement:** Data sharing is not applicable to this article.

**Conflicts of Interest:** The authors declare no conflict of interest.

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
