# Peer review of "Developing Techniques for Closed-Loop-Recycling Soda-Lime Glass Fines through Robotic Deposition"

_arts_

Round 1
Reviewer 1 Report
The article has a great potential for artistic development if focus on Robotic Deposition.
There is lack of information on the state of art. The author says that robotic deposition has been explored in one previous research, but does not mention any reference to the previous project. If there are studies made on this area they should be referred.
On point 3 the author speaks about Danish market, it is the first time this is mention. It is not clear if the geographical area is circumscribed to this location. If so, why?
The point 5 – testing material composition is well developed it would be important to have an image of shape before firing.
Point 6 - artistic application - it is a succession of failed outcomes.
There is no specification of the aneling schedule, and it should have a better scientific development, as the artistic does not have any. If there is no compatibility between these materials and the test pieces break, how can the artist use this technique in the artistic conceptualization? The need to have the tests made on the mentioned techniques is meritorious and necessary, but there should be some positive results. The article should focus more with the good outcomes.
I do not agree when the author says “ to adjust the coefficient of expansion for the two material do not seem comprehensible within this research -if at all”
Compatible test should be made before.
What is the relevance to have a material that the author says it is for art application and in the end this material cannot be used in the techniques mention because the materials are not compatible?
If it is not possible to combine these materials in the hot process, they should be tested in a could method with several adhesives.
In 6.3 there is a good result. The article should focus more in this.
Why the casting technique was not used ? why printed RSGF was not combined with casting RSGF?
The sustainable development (7) and the method (8) should have a deeper elaboration.
In the attachment I add more observations

Minor editing of English language required
Reviewer 2 Report
In my opinion, the article and the research are interesting however not fully carried out. The authors do not consider the possibilities that are available and described in the literature to avoid or reduce the problem of incompatibility between two materials.
The authors only mention the CTE (coefficient of thermal expansion) when comparing the stability of the materials they use in their research.
This is indeed an important parameter, but usually not if we want to consider the stability of recycling materials at high temperatures, as CTE constantly change during the process, and not uniformly with heating, not linearly with temperature and the materials have different shrinkage stages during the heating and cooling process.
At high temperatures, as is generally known, glass can undergo various physical and chemical changes that can affect its properties and performance and usually creating a strong bond with other materials, because its wetting characteristic. A COE of materials can be different only if there is not created a strong bond between materials during the cooling process.
Therefore, some study showed that to avoid glass wetting in high temperatures and brake bond between materials, it is possibility to use separators. Additionally, to prevent incompatibility as much as possible during cooling process, it should be considering annealing programs for both materials in the same time.
In my opinion, the empirical studies described in the article only show the interesting idea of using recycled glass but do not show an elaborate method of how to combine the two materials to have better control over the compatibility. I think research needs more experiments in this direction.
I have feeling that the first two sentences in Chapter 8 should be rephrased as they sound confusing:
„The technical issues discovered in this research are discouraging further research into combining 3D printed RSGF with blown container glass and/or blown RSGF. The formal and aesthetic findings of the research are showing promising results with regard to 3D printing the RSGF as well as with regard to melting and blowing the RSGF”.
Round 2
Reviewer 1 Report
I understand what is struggling for funding to make the research. The compatible is critical for the success of this project. Of course, it would not be possible to make this investigation now for the paper, but when writing this article without this compatibility tests the author should altered the methodology. Different approach needs to be done in the writing and other factor should be taking into consideration.
The state of the art should be better improved. The author says this research is for artists to use this material. In that case we need to understand what was done on the previous research of Thomsen et al.
Point 3 . 3. The circular model for recycled container glass
The author should explain better the Experimental Procedure
recycled soda-lime glass fines (RSGF) – how fine is this? How do you prepare the material for the printer? A photography of the material will be interesting.
In this section should be mentioned all the steps an artist’s needs to make in preparing the glass (RSGF) and the design of the piece.
Point 6 . the author add more information is better but not organized, and it is confusing. There are repeated titles and it would be interesting to develop this points better
6.3.1. Technical material analysis of RSGF combined with blown RSGF
“ we have not yet been able to locate a lab that has the capacity to conduct a test for us.”, this is understandable, but you should not finish the sentence like this !
6.4. 3D printing options for RSGF and addition of ceramic stains
- explain the ceramic composition. What did you use?
6.4.1. Technical material analysis of RSGF combined with blown RSGF
What is the difference of point 6.3.1?
It is confusing….
6.3.2. Aesthetic analysis of 3D printed RSGF combined with blown RSGF
6.4.2. Aesthetic analysis of 3D printed RSGF combined with blown RSGF
The titles should be different….
Please confirm the numeration of the figures.
In the previous comment it was suggested that “The sustainable development” (7) and the method (8) should have a deeper elaboration. The author did this on point 8, but not in point 7.
Minor editing of English language required
Author Response
"Please see the attachment."

Round 3
Reviewer 1 Report
The changes made by the author contribute to a better quality of the article.
I wish the author every success in obtaining grants to develop this investigation and hope in the future to read more information on the development of this research.
Author Response
We would like, once again, to express our sincere gratitude for your thorough, constructive and detailed feedback. We feel blessed to have received such a professional response, and we believe our work has improved significantly by our attempts to respond to your feedback. Thank you for your time and consideration it is highly appreciated! And thank you also for your kind remarks. :)